# A bacterial riboswitch class for the thiamin precursor HMP-PP employs a terminator-embedded aptamer

Ruben M Atilho[1†], Gayan Mirihana Arachchilage[2†], Etienne B Greenlee[3], Kirsten M Knecht[1], Ronald R Breaker[1,3]*

[1]Department of Molecular Biophysics and Biochemistry, Yale University, New Haven, United States; [2]Howard Hughes Medical Institute, Yale University, New Haven, United States; [3]Department of Molecular, Cellular and Developmental Biology, Yale University, New Haven, United States

**Abstract** We recently implemented a bioinformatics pipeline that can uncover novel, but rare, riboswitch candidates as well as other noncoding RNA structures in bacteria. A prominent candidate revealed by our initial search efforts was called the '*thiS* motif' because of its frequent association with a gene coding for the ThiS protein, which delivers sulfur to form the thiazole moiety of the thiamin precursor HET-P. In the current report, we describe biochemical and genetic data demonstrating that *thiS* motif RNAs function as sensors of the thiamin precursor HMP-PP, which is fused with HET-P ultimately to form the final active coenzyme thiamin pyrophosphate (TPP). HMP-PP riboswitches exhibit a distinctive architecture wherein an unusually small ligand-sensing aptamer is almost entirely embedded within an otherwise classic intrinsic transcription terminator stem. This arrangement yields remarkably compact genetic switches that bacteria use to tune the levels of thiamin precursors during the biosynthesis of this universally distributed coenzyme.
DOI: https://doi.org/10.7554/eLife.45210.001

*For correspondence:
ronald.breaker@yale.edu

[†]These authors contributed equally to this work

Competing interests: The authors declare that no competing interests exist.

## Introduction

Approximately 40 distinct riboswitch classes that regulate gene expression in various bacterial species have been experimentally validated to date (*McCown et al., 2017*; *Serganov and Nudler, 2013*; *Sherwood and Henkin, 2016*; *Breaker, 2011*). Based on the abundances and distributions of these known riboswitch classes, it has been proposed that many thousands of additional riboswitch classes remain to be discovered in the eubacterial domain of life (*Ames and Breaker, 2010*; *Breaker, 2012*; *McCown et al., 2017*). The collection of known riboswitch classes largely sense compounds or ions that are of fundamental importance to organisms from all three domains of life, and these ligands also exhibit a bias in favor of compounds (enzyme cofactors, RNA nucleotides and their precursors or derivatives) that are predicted to be of ancient origin (*Breaker, 2012*; *McCown et al., 2017*; *Nelson and Breaker, 2017*). If these trends hold, it seems likely that numerous additional riboswitch classes that regulate fundamental biological processes remain to be discovered. Unfortunately, the vast majority of these undiscovered riboswitch classes are predicted to be exceedingly rare, and this characteristic is likely to cause difficulties for researchers who seek to identify them.

To address this challenge, we developed a computational pipeline that first identifies the regions of bacterial genomes that are most likely to serve as transcription templates for structured noncoding RNAs (ncRNAs), and then uses comparative sequence and structural analyses to identify novel candidate RNA motifs (*Stav et al., 2019*). Specifically, this approach examines only the putative

**eLife digest** Many bacteria use small genetic devices called riboswitches to sense molecules that are essential for life and regulate the genes necessary to make, break or move these molecules. Riboswitches are made of molecules of RNA and appear to have ancient origins that predate the evolution of bacteria and other lifeforms made of cells. Inside modern bacteria, chunks of DNA in the genome provide the instructions to make riboswitches and around 40 different types of riboswitch have been identified so far. However, it has been proposed that the instructions for thousands more riboswitches may remain hidden in the DNA of bacteria.

All of the currently known riboswitches contain a region called an aptamer that binds to a target molecule. This binding causes another structure in the riboswitch RNA to switch a specific gene on or off. For example, the aptamer binding might cause a hairpin-like structure called a terminator to form, which stops a gene being used to make new RNA molecules.

In 2019 a team of researchers reported using a computational approach to identify new riboswitches in bacteria. This approach identified many different chunks of DNA that might code for a riboswitch, including one known as the *thiS* motif. This potential new riboswitch appeared to be associated with a gene that encodes a protein required to make a vitamin called thiamin (also known as vitamin B$_1$).

To test the new computational approach, Atilho et al. including several of the researchers involved in the earlier work used genetic and biochemical techniques to study the *thiS* motif. The experiments revealed that the motif binds to a molecule called HMP-PP, which bacteria use to make thiamin. Unexpectedly, the aptamer of the riboswitch was nested within a terminator, rather than being a separate entity.

The findings of Atilho et al. reveal that riboswitches can be even more compact than previously thought. Furthermore, these findings reveal new insights into how bacteria use riboswitches to manage their vitamin levels. In the future it may be possible to develop drugs that target such riboswitches to starve bacteria of these essential molecules.

DOI: https://doi.org/10.7554/eLife.45210.002

noncoding regions of a given sequenced bacterial genome, and evaluates each intergenic region (IGR) based on two parameters: (*i*) percent guanosine and cytidine (GC) nucleotide content and (*ii*) length in nucleotides. For many bacteria, structured ncRNAs are GC-rich compared to other regions of the genome (*Klein et al., 2002*; *Schattner, 2002*), and the IGRs that serve as synthesis templates for these ncRNAs tend to be much longer than typical bacterial IGRs that contain only an RNA polymerase promoter and/or a protein-specific regulatory domain. Our current bioinformatics pipeline is based on earlier implementations of this search strategy that were used to discover several novel structured ncRNA motifs (*Meyer et al., 2009*), including the SAM-V riboswitch class (*Poiata et al., 2009*).

Our updated computational pipeline was employed to comprehensively examine the genomes of five bacterial species, which revealed the existence of as many as 70 novel genetic elements, including 30 candidate ncRNA motifs (*Stav et al., 2019*). Of the eight candidate riboswitch classes uncovered in this search, the most promising was called the '*thiS* motif' (*Figure 1A*) because representatives most commonly reside immediately upstream of *thiS* genes, which code for a protein that delivers sulfur to the pathway for the production of the thiamin biosynthetic intermediate 5-(2-hydroxyethyl)−4-methylthiazole phosphate (HET-P) (*Begley et al., 2012*). Other genes associated with *thiS* motif RNAs appear to code for proteins that participate in the production of HET-P or its fusion to 4-amino-5-hydroxymethyl-2-methylpyrimidine diphosphate (HMP-PP), to ultimately produce the bioactive coenzyme thiamin pyrophosphate (TPP) (*Jurgenson et al., 2009*).

An additional clue regarding the function of *thiS* motif RNAs was derived from the fact that this novel RNA structure occasionally resides in tandem with TPP riboswitches (*Stav et al., 2019*). Tandem arrangements of other riboswitch classes have been shown to function as two-input Boolean logic gates (*Sudarsan et al., 2006*; *Stoddard and Batey, 2006*; *Lee et al., 2010*; *Sherlock et al., 2018*), suggesting that the genes associated with tandem arrangements of TPP riboswitches and *thiS* motif RNAs likely respond to concentration changes of two distinct ligands. Together, these

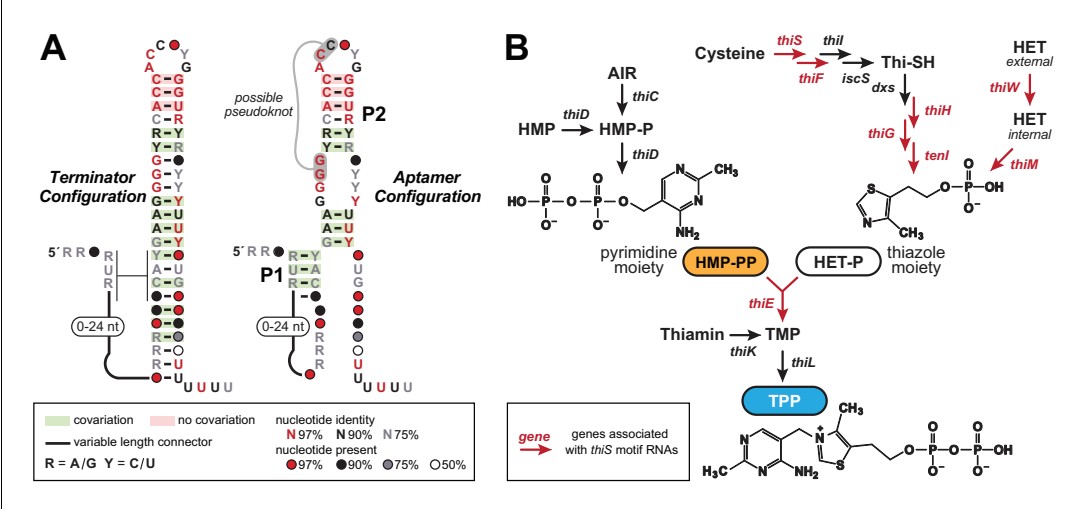

**Figure 1.** The *thiS* motif consensus models and gene associations. (**A**) Conserved nucleotide sequences and two possible secondary structure models based on 400 representatives of *thiS* motif RNAs. (Left) The terminator configuration exhibits features (strong stem followed by a run of U nucleotides) that are consistent with a typical bacterial intrinsic transcription terminator stem ('OFF' state). (Right) Another secondary structure model ('ON' state) for *thiS* motif RNAs, consistent with the pattern of conserved base-pairing potential, involves the formation of a distinct P1 stem and a possible pseudoknot, whereas the base-pairing of P2 is identical to the tip of the terminator stem configuration. (**B**) Typical biosynthetic pathways for the pyrimidine and thiazole moieties of thiamin, and their eventual conversion into the enzyme cofactor thiamin pyrophosphate (TPP). Genes coding for the enzymes involved in HET transport and catalysis of various biosynthetic steps are annotated with colors that reflect their association with *thiS* motif representatives. Note that a carboxylated derivative of HET-P, called cHET-P, has been proposed to be the main thiazole-based biosynthetic intermediate for TPP in some bacteria (*Begley et al., 2012*).

DOI: https://doi.org/10.7554/eLife.45210.003

observations strongly indicate that *thiS* motif RNAs function as riboswitches that respond to a biochemical intermediate of the TPP biosynthetic pathway.

Previously (*Stav et al., 2019*), we created a riboswitch-reporter fusion construct by joining a *thiS* motif RNA representative to a β-galactosidase gene. Using this construct, we observed robust gene expression in host *Bacillus subtilis* cells with a deleted *thiS* gene (Δ*thiS*). This same reporter-fusion construct yields no reporter gene expression in host *B. subtilis* cells that naturally carry a *thiS* gene. These findings were consistent with our hypothesis that *thiS* motif RNAs function as riboswitches, but the precise ligand sensed by the unusual architecture of this RNA remained unknown.

In the current report, we describe a series of bioinformatic, genetic and biochemical analyses that provide conclusive evidence that the ligand for *thiS* motif RNAs is the TPP biosynthetic intermediate HMP-PP. Furthermore, our findings demonstrate that this unusually small riboswitch employs a distinct architecture to regulate RNA transcription termination. This regulatory RNA provides cells with an efficient mechanism to balance the production of two key biosynthetic intermediates, HMP-PP and HET-P, which are then fused to make the essential coenzyme TPP.

## Results and discussion

### The unusual architecture and genetic distribution of the *thiS* riboswitch candidate

Most experimentally validated riboswitch classes are composed of distinct, but partially overlapping aptamer and expression platform domains (*Barrick and Breaker, 2007*; *Breaker, 2012*). In contrast, the *thiS* motif exhibits an unusual arrangement wherein the predominant secondary structure, derived by thermodynamic modeling, is an extended hairpin structure that maximizes conventional Watson/Crick base pairing (*Figure 1A*, left). This structure exhibits all the features characteristic of bacterial intrinsic terminator stems, including an uninterrupted and strong base-paired stem followed by a run of six or more uridine (U) residues (*Wilson and von Hippel, 1995*; *Yarnell and*

*Roberts, 1999*). As a result, we concluded that transcription termination was certain to be a major function of *thiS* motif RNAs.

We frequently encounter simple terminator stems when using bioinformatics search algorithms to identify novel ncRNA candidates, and we have now begun to assign predicted terminator function to such motifs and then quickly move on to examine other more promising ncRNA candidates. When evaluating the *thiS* candidate, however, we noted four features that suggested this terminator stem was peculiar. First, the loop of the terminator hairpin is abnormally well conserved compared to the loop sequences of more typical terminator stems, which are usually irrelevant to the mechanism of transcription termination. This unique terminator element is found in species from two phyla and from several classes within Firmicutes (*Supplementary file 1*, *Supplementary file 2*). Second, another consensus structural model was also consistent with the comparative sequence analysis data (*Figure 1A*, right). This architecture, including a possible pseudoknot and two major base-paired stems, disrupts the contiguous terminator stem near the run of U residues, suggesting that mutually exclusive and competing structures and functions might exist for *thiS* motif RNAs. Third, these apparently specialized terminator stems associate exclusively with genes related to the biosynthesis and utilization of HET-P, which is a precursor of the coenzyme TPP (*Figure 1B*). These sequence, structure, and genomic distribution characteristics suggested to us that each *thiS* motif RNA might function as a compact ligand sensor and regulator of TPP coenzyme biosynthesis.

A fourth feature of *thiS* motif RNAs is that approximately 30% of the known representatives reside immediately downstream of riboswitches that sense and respond to TPP (*Stav et al., 2019*). Each associated TPP riboswitch appears to use a terminator stem as an expression platform. The tandem arrangement of two terminator stems for a single riboswitch aptamer would be unprecedented, and so this observation also supported our hypothesis that *thiS* motif RNAs represent an unusual form of riboswitch. Due to these tandem arrangements, we speculated that the natural ligand for this riboswitch candidate would not be TPP. Rather, it seemed more likely that the ligand would be a major precursor of TPP (either HET-P or HMP-PP), and that each tandem TPP-*thiS* motif system would function as a two-input Boolean logic gate (*Sudarsan et al., 2006*; *Stoddard and Batey, 2006*; *Lee et al., 2010*; *Sherlock et al., 2018*) to regulate HET-P production in response to the cellular concentrations of both TPP and one of its biosynthetic precursors.

Furthermore, if the two secondary structure states called the 'terminator configuration' and the 'aptamer configuration' (*Figure 1A*) represent the riboswitch 'OFF' and 'ON' configurations, respectively, then ligand binding by the aptamer in *thiS* motif RNAs is expected to activate gene expression. The vast majority of riboswitches that sense metabolites and control biosynthetic pathways are OFF switches, and thus the accumulation of the metabolite ligand results in reduced expression of the protein products that otherwise would make (or import) more of the desired metabolite. Although *thiS* motif representatives associate with genes for the production of HET-P (*Figure 1B*), it does not make sense that HET-P would turn on its own production when it is already abundant. Finally, we recognized that the *thiE* gene (coding for thiamin-phosphate synthase) is occasionally associated with *thiS* motif representatives, suggesting that the RNA motif determines if cellular conditions are suitable to couple the two key precursors of thiamin monophosphate (TMP) to eventually yield TPP. Taken together, all these bioinformatic observations are consistent with the hypothesis that *thiS* motif RNAs function as ON riboswitches for the TPP precursor HMP-PP.

## A *thiS* riboswitch-reporter fusion construct is activated by the addition of HMP to cells

As an initial test of our hypothesis that HMP-PP is the ligand for *thiS* motif RNAs, we employed a chromosomally-integrated reporter construct (*Stav et al., 2019*) carrying a β-galactosidase gene fused downstream of nucleotides encompassing the *thiS* motif from *Clostridium* species Maddingley (*Rosewarne et al., 2013*) (*Figure 2A*). Transformed *B. subtilis* cells carrying a transcriptional fusion of the wild-type (WT) riboswitch to a *lacZ* reporter gene, which was integrated into the *amyE* locus, exhibited no β-galactosidase activity in response to HMP added to rich (LB) liquid culture media (*Figure 2B*). Perhaps the natural suppression of TPP biosynthesis when cells have sufficient amounts of this coenzyme precludes the formation of excess HMP-PP in this surrogate organism. Specifically, if translation of the *thiD* gene coding for HMP/HMP-P kinase is largely suppressed under normal cellular conditions, the externally supplied HMP cannot be phosphorylated to generate HMP-PP.

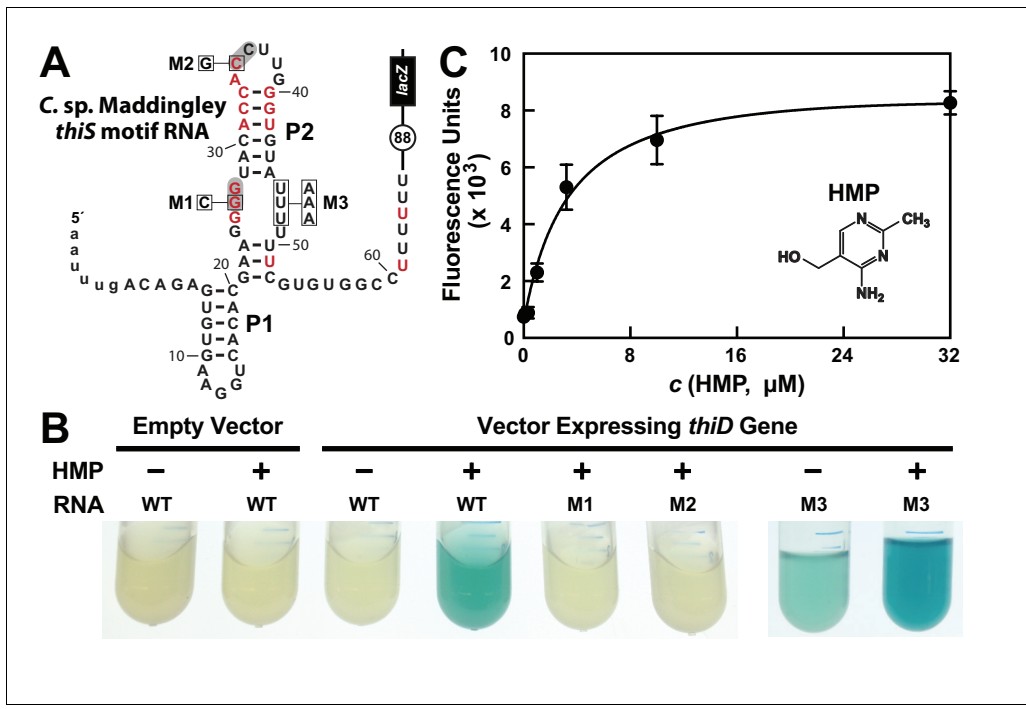

**Figure 2.** HMP-PP triggers reporter gene expression. (**A**) Sequence and predicted secondary structure of the WT *thiS* motif RNA associated with the *thiS* gene of *C.* sp. Maddingley fused to a β-galactosidase reporter gene (*lacZ*). The structural model is depicted in its presumed 'ON' state, where the shaded nucleotides highlight possible base-pairs of the predicted pseudoknot. The encircled number represents the additional nucleotides linking the end of the intrinsic terminator element (6 U nucleotides) and the coding region joined to the *lacZ* reporter gene sequence. Red nucleotides are >97% conserved as depicted in the *thiS* consensus model (*Figure 1A*). The nucleotides are numbered beginning with the predicted natural transcription start site, and the lowercase letters at the 5' end identify additional nucleotides included in the reporter-fusion construct. Boxed nucleotides identify positions that are altered in the mutant constructs indicated. (**B**) Photos of reporter gene assay liquid cultures of *B. subtilis* cells carrying the protein expression plasmid pDG148-Stu either lacking (empty vector) or carrying a native *thiD* gene. Reporter constructs carrying the WT riboswitch construct or mutations M1 through M3 as depicted in A were grown in LB either without (–) or with (+) HMP supplementation. (**C**) HMP concentration dependent reporter gene expression of WT *B. subtilis* cells carrying the riboswitch-*lacZ* fusion construct depicted in A. Cells were grown in minimal (GMM) liquid medium supplemented with various concentrations of HMP as indicated. Data points are the average of three measurements and are representative of experiments performed on multiple days. Error bars indicate the standard deviation of the measurements, which when not visible are smaller than the corresponding data points.

DOI: https://doi.org/10.7554/eLife.45210.004

The following source data and figure supplement are available for figure 2:

**Source data 1.** Fluorescence values for reporter gene expression data for *Figure 2C*.
DOI: https://doi.org/10.7554/eLife.45210.006
**Figure supplement 1.** The effects of thiamin biosynthetic pathway knockouts on expression of a *thiS* riboswitch-reporter construct.
DOI: https://doi.org/10.7554/eLife.45210.005

In contrast, robust β-galactosidase activity is evident in cells co-transformed with the reporter construct and a vector overexpressing the ThiD protein, but only when growth media is supplemented with HMP (*Figure 2B*). Reporter gene expression levels are dependent on the concentration of HMP added to the culture medium (*Figure 2C*), suggesting that cells take up HMP and use the *thiD* gene product to add two phosphates to generate excess HMP-PP, which activates *lacZ* reporter expression by binding to the riboswitch.

Furthermore, the characteristics of a series of mutant riboswitch-reporter constructs examined in *B. subtilis* cells likewise indicates that *thiS* motif RNAs function as riboswitch aptamers for HMP-PP. Constructs carrying mutations M1 (G26C) or M2 (C35G) (*Figure 2A*), which alter strictly-conserved

nucleotides that are predicted to be part of the pseudoknot structure of the aptamer (*Figure 1A*), are not activated by HMP addition (*Figure 2C*). Importantly, construct M3 (U46A, U47A, U48A) exhibits a detectable level of reporter gene expression in LB media alone, and this expression is further enhanced by the addition of HMP. This result suggests that these nucleotides are involved in forming a strong terminator stem, but that the identities of the nucleotides at these positions are not critical for ligand binding by the putative HMP-PP aptamer.

The WT riboswitch-reporter construct (*Figure 2A*) was also used to assess gene regulation in response to differences in growth media, and to several genetic disruptions of the TPP biosynthetic pathway (*Figure 2—figure supplement 1*). *B. subtilis* cells grown in LB medium are expected to suppress the genes needed to produce both the pyrimidine moiety HMP-PP and the thiazole moiety HET-P. Thus, as expected, expression of the reporter gene fused to the putative HMP-PP riboswitch is off, regardless of the genetic background tested. Likewise, WT cells grown in minimal (GMM) medium also do not express the reporter gene, presumably because HMP-PP does not accumulate due to its rapid and efficient conversion to the final product, TPP. By contrast, ΔthiS and ΔthiE cells (carrying genetic disruptions of the *thiS* and *thiE* genes, respectively) exhibit high levels of reporter gene expression (*Figure 2—figure supplement 1*). These two genetic knock-out strains are predicted to accumulate HMP-PP because they either lack the protein (ThiS) that initiates the production of HET-P, or they lack the protein (ThiE) that fuses HMP-PP to HET-P even when HET-P is available.

Additional support for the hypothesis that HMP-PP is the ligand, rather than HMP-P, is provided by the level of expression of the reporter construct when present in host cells carrying a deletion of the gene (ΔthiD) required to phosphorylate HMP-P to make HMP-PP. Reporter gene expression is not observed in ΔthiD cells (*Figure 2—figure supplement 1C*), despite the fact that these cells should accumulate HMP-P. Overall, these genetic results strongly indicate that HMP-PP is the ligand for a riboswitch class represented by *thiS* motif RNAs that turns on gene expression when this ligand is abundant.

## HMP-PP suppresses intrinsic transcription termination in vitro

The two possible architectures of the *thiS* motif RNA from *C.* sp. Maddingley (*Figure 3A*), also observed in other members of this candidate riboswitch class (*Figure 1A*), suggest a mechanism for gene regulation involving the mutually exclusive formation of an intrinsic terminator stem and its competing ligand-bound aptamer state. Such transcription control mechanisms for riboswitches have been experimentally validated in the past (e.g. *McDaniel et al., 2003*; *Mironov et al., 2002*; *Sudarsan et al., 2003*; *Wickiser et al., 2005a*) by using single-round in vitro transcription assays (*Landick et al., 1996*) to reveal ligand-dependent modulation of RNA transcript lengths. Therefore, we sought to examine the proposed riboswitch mechanism by conducting in vitro transcription reactions with the expected natural ligand, HMP-PP.

Unfortunately, our use of such transcription assays was made more difficult, because HMP-PP is not commercially available and is known to be relatively unstable (*Hanes et al., 2007*). Therefore, we freshly generated HMP-PP from HMP and ATP enzymatically (*Figure 3B*) by using recombinantly produced ThiD protein. HMP-PP production as described previously (*Hanes et al., 2007*) was confirmed by mass spectrum analysis (*Figure 3C* and *Figure 3—figure supplement 1*). The enzymatic reactions were deproteinized by filtration, and the resulting mixtures containing HMP-PP were added, without further purification, to various assays as described for each experiment.

The DNA template for the WT *thiS* motif RNA construct (*Figure 3A*) yields only ~14% full-length transcript when transcription reactions are conducted in the presence of HMP alone. In contrast, nearly 40% of the transcripts are full-length (FL) when transcription reactions are conducted in the presence of enzyme-prepared HMP-PP (*Figure 3D*). Furthermore, mutations M1 and M2, which disrupt possible pseudoknot formation by the aptamer and which eliminate gene expression in vivo (*Figure 2C*), fail to produce more FL transcripts when HMP-PP is present. These results suggest that HMP-PP binding induces RNA polymerase to transcribe past the intrinsic terminator element to yield fewer terminated (T) products and a greater proportion of FL RNAs.

From the assay data presented, this switching effect is incomplete in vitro, as evident by the fact that construct M4, which carries three nucleotide changes that disrupt the otherwise perfect base pairing of the terminator stem, yields nearly 100% 'FL' transcripts. Construct M5 yields transcripts that approximate the length of 'T' RNAs, which are generated naturally by termination due to the

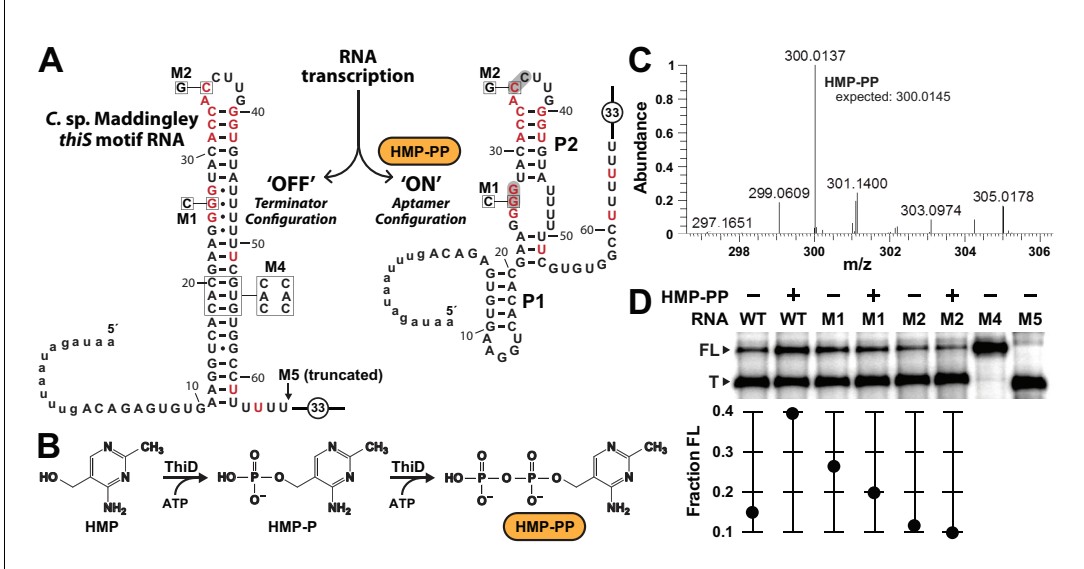

**Figure 3.** Pyrophosphorylated HMP triggers riboswitch-mediated transcription elongation in vitro. (A) Sequence and secondary structure models for the 'OFF' and 'ON' states of an HMP-PP riboswitch construct from *C.* sp. Maddingley used for transcription termination assays. In the absence of ligand ('OFF' state, left), an intrinsic transcription terminator stem is expected to form and cause transcription termination near nucleotide number 66. In the ligand-bound structure ('ON' state, right), a distinct P1 stem and a P2 stem (partial formation of the terminator) are expected to form, thereby blocking formation of the complete terminator stem and leading to transcription read-through. (B) Pathway for the enzymatic pyrophosphorylation of HMP using 4-amino-5-hydroxymethyl-2-methylpyrimidine phosphate kinase (ThiD), carried out as previously described (*Hanes et al., 2007*). (C) Plots of the mass-to-charge ratios (m/z) of HMP and HMP-PP prepared by enzymatic pyrophosphorylation. Relative abundance (peak heights) are normalized to that of HMP-PP. (D) (Top) PAGE analysis of single-round transcription termination assays of WT and various mutant HMP-PP riboswitch constructs in the absence or presence of ligand. 'FL' and 'T' denote the full length and terminated transcripts, respectively. Transcription reactions in lanes denoted '(–) HMP-PP' were supplemented with the processed reaction mixture including all the reagents (except ThiD protein) required for the preparation of HMP-PP. Thus, these lanes include ~0.5 mM HMP. (Bottom) Values for the fraction of full-length RNA transcript relative to the total transcription yield are plotted for each reaction.

DOI: https://doi.org/10.7554/eLife.45210.007

The following figure supplement is available for figure 3:

**Figure supplement 1.** Mass spectrum analyses of the HMP-PP enzymatic biosynthesis reactions before (top) and after (bottom) the addition of ThiD protein.

DOI: https://doi.org/10.7554/eLife.45210.008

action of the intrinsic terminator stem. Incomplete switching by terminator-regulating riboswitches is typical for such assays conducted in vitro (e.g. *McDaniel et al., 2003*; *Mironov et al., 2002*; *Sherlock et al., 2018*; *Sudarsan et al., 2003*; *Wickiser et al., 2005a*). However, it seems likely that a larger dynamic range for transcription control is exploited by this riboswitch in cells, as evident by the robust differences in gene expression for the nearly identical reporter constructs used in this study (*Figure 2* and *Figure 2—figure supplement 1*).

## Biochemical evidence for direct binding of HMP-PP by RNA aptamers

The unique architecture of *thiS* motif RNAs also served as an obstacle for evaluating the ability of these RNAs to directly bind a ligand. We frequently employ in-line probing assays (*Soukup and Breaker, 1999*; *Regulski and Breaker, 2008*) to determine if RNAs undergo structural changes in response to ligand binding. For *thiS* motif RNAs, the terminator stem is expected to dominate over the aptamer configuration (*Figure 1A*) simply due to the different number of base-pairs present in each structural state. Therefore, under thermodynamic equilibrium conditions typically experienced by RNAs subjected to in-line probing reactions, a full-length construct is not expected to bind HMP-PP because the RNA will always favor the terminator configuration ('OFF' state).

To address this problem, we reasoned that a shorter construct that weakens the terminator stem might permit the aptamer configuration to be adopted. Such constructs are also likely to better represent the structures naturally adopted by the riboswitch during transcription. An RNA polymerase

paused at the run of U nucleotides at the end of the intrinsic terminator element will sequester ~12 nucleotides of the transcript within the protein structure (*Monforte et al., 1990*; *Komissarova and Kashlev, 1998*; *Vassylyev et al., 2002*). Therefore, only the first 50 to 54 nucleotides of the nascent transcript of the natural *C.* sp. Maddingley *thiS* motif should be exposed if the RNA polymerase complex is paused within the run of U nucleotides of the intrinsic terminator stem.

A series of four *thiS* motif RNA constructs was prepared to investigate the hypothesis that shorter constructs might permit aptamer formation by weakening the terminator stem. The longest RNA construct carries the full terminator stem (66 *thiS*), and three progressively truncated variants (54 *thiS*, 53 *thiS* and 52 *thiS*) represent only the RNA regions that would be exposed when RNA polymerase is stalled at various locations within the run of U nucleotides (*Figure 4A*). In-line probing assays (*Figure 4B*) reveal that 66 *thiS* indeed exclusively forms the terminator stem, as evident by robust spontaneous RNA cleavage in the unstructured regions near the 5′ terminus through nucleotide position 9, and by the unstructured loop region including nucleotides 35 through 39. This loop

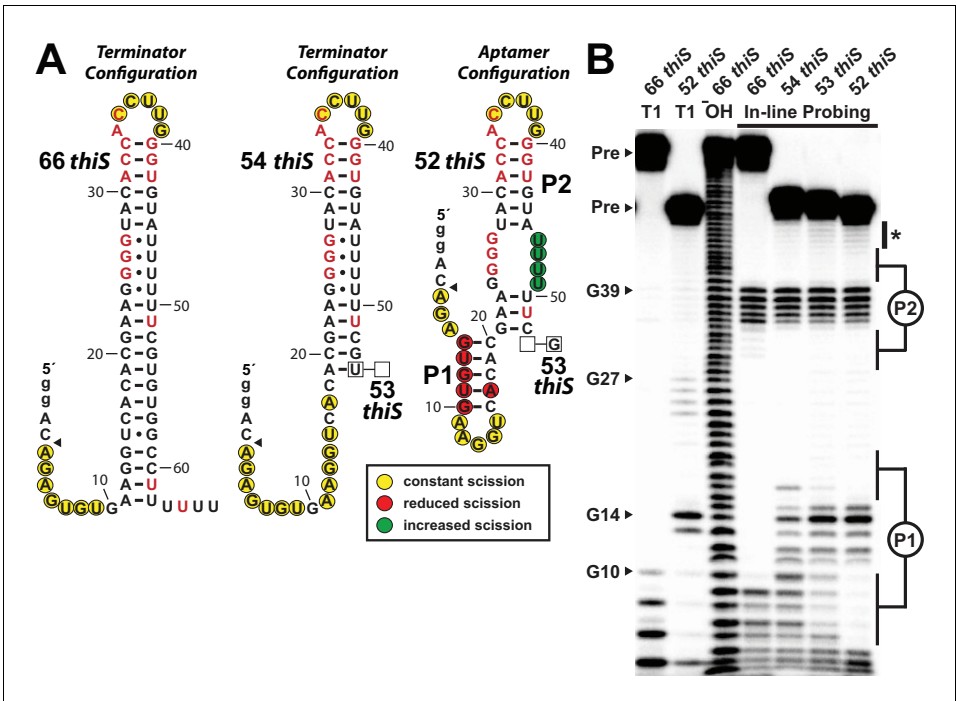

**Figure 4.** Substantial conformational rearrangement of an HMP-PP riboswitch is brought about by single-nucleotide differences at the 3′ terminus. (**A**) Sequences and secondary structure models of RNA constructs used to assess conformational determination between terminator and aptamer structures. Nucleotide differences between constructs are boxed, and empty boxes indicate the nucleotide is absent. The secondary structure models were predicted based on the in-line probing data derived from B. Note that the 53 *thiS* RNA secondary structure model adopts both the terminator (predominant form for 54 *thiS*) and aptamer (predominant form for 52 *thiS*) conformations with near equal probability. Yellow circles identify nucleotide linkages that undergo robust spontaneous strand scission. Some remain unstructured regardless of the length of the construct, suggesting these nucleotides always remain unstructured. Red circles identify linkages that exhibit reduced spontaneous scission compared to larger constructs, suggesting the corresponding nucleotides become more structured in the 52 *thiS* construct. Green circles identify linkages undergoing increased strand scission, suggesting these nucleotides become less structured in 52 *thiS* construct. The arrowhead identifies the start of the interpretable data in B. (**B**) PAGE analysis of 5′ $^{32}$P-radiolabeled RNA constructs subjected to in-line probing analysis in the absence of HMP-PP. T1 and $^-$OH indicate RNase T1 partial digestion (cleavage after G nucleotides) and alkaline-mediated partial digestion (cleaves after each nucleotide), respectively. Bands corresponding to precursor RNA (Pre) and enzymatic cleavage after certain G residues are annotated. The asterisk identifies bands corresponding to nucleotides 46–49 of 52 *thiS*, which appear to increase in intensity compared to the 66 *thiS* construct.
DOI: https://doi.org/10.7554/eLife.45210.009

region remains unstructured with all constructs tested, demonstrating that the upper base-paired portion is common to all structural configurations.

Construct 54 *thiS* appears to retain all base-pairs of the terminator stem, except those that are disrupted by the deletion of the 3′-terminal nucleotides from position 54 and beyond (*Figure 4B*). In stark contrast, construct 52 *thiS* adopts a configuration that matches the consensus model for the aptamer configuration, including the formation of stems P1 and P2 (*Figure 1A*). Intriguingly, this shortest construct exhibits modest evidence (*Figure 4B*, asterisk) of a weakening of base-pairing within the series of G-U wobble interactions formed between nucleotides 24 to 27 and nucleotides 46 to 49. This structural flexibility might permit the formation of a pseudoknot or some other tertiary interactions between these highly-conserved G nucleotides and pyrimidines in the loop region upon ligand binding. Also noteworthy is the fact that the intermediate length construct, 53 *thiS*, exhibits an in-line probing pattern consistent with the formation of both the terminator configuration and the aptamer configuration. These results suggest that the genetic decision whether to form the terminator stem configuration or the ligand-bound aptamer configuration is made within a very narrow window of transcription progression.

Given the ability of the 52 *thiS* construct to exclusively adopt the aptamer configuration, we used this RNA to seek additional evidence for direct binding of HMP-PP by the riboswitch. However, this construct still failed to exhibit ligand-induced changes in the banding pattern resulting from in-line probing reactions. We speculate that nucleotide positions 24–27 and positions 46–49 form base-pairs that trap this portion of the motif in its terminator configuration. To further favor the formation of the desired aptamer configuration, the M3 version of the 52 *thiS* construct was made (*Figure 5A*). The same M3 mutations, made in the context of the full-length construct, earlier were observed to retain ligand responsiveness in riboswitch-reporter assays in vivo (*Figure 2B*).

The 52 *thiS* M3 construct indeed exhibits structural modulation upon introduction of HMP-PP (*Figure 5B*). Although the main base-paired regions P1 and P2 remain unchanged by HMP-PP addition, nucleotides involved in forming the putative pseudoknot (sites 1 and 2), and other nucleotides in the loop of P2 (site 3) appear to become more structured. Additional mutations were introduced into construct 52 *thiS* M3 to evaluate the effects of mutations known or expected to disrupt gene control function. Specifically, previous mutations M1 or M2 (*Figure 2A*), or M6 (*Figure 5—figure supplement 1A*) were introduced to create constructs, termed M7, M8 and M9, which carry alterations to highly-conserved nucleotides that presumably disrupt pseudoknot formation. These additional mutations eliminate structural modulation by HMP-PP (*Figure 5—figure supplement 1B*), as would be expected if these highly-conserved sequence and structural features are critical for riboswitch aptamer function.

We suspect that the ligand concentration is insufficient to saturate all RNAs in the sample. Regardless, by quantifying the band intensities at the three sites of structural modulation and by assuming that these values could become zero (or background) upon ligand binding, we generated a partial HMP-PP binding curve for 52 *thiS* M3 (*Figure 5C*), which is consistent with a 1-to-1 interaction between the ligand and the RNA aptamer. Given our use of enzymatically prepared HMP-PP samples, we cannot use this data to precisely determine the dissociation constant ($K_D$) for this interaction. However, if we assume that all HMP was enzymatically converted into HMP-PP, this product was fully recovered after removal of protein by filtration, and that there was no degradation over the time frame of the assays, the $K_D$ value cannot be greater than 500 µM for this variant aptamer construct.

A series of shortened constructs carrying the M3 mutations (*Figure 5—figure supplement 2A*) recapitulate the transition between the terminator and aptamer configurations as originally observed for the series of WT truncated RNAs (*Figure 4*). Importantly, the 53 *thiS* M3 construct exhibits evidence of both binding to HMP-PP and of structural switching to favor the aptamer configuration ('ON' state) (*Figure 5—figure supplement 2B*). This finding suggests that it is necessary to prevent the formation of G-U wobble base-pairs between nucleotide positions 25–27 and positions 46–48 for ligand binding to be observed by using in-line probing assays. The time scale of in-line probing reactions (24 to 48 hr) is much longer than the time scale for the natural genetic decision to take place (probably a few seconds). Therefore our biochemical assays permit the RNAs to reach thermodynamic equilibrium (*Wickiser et al., 2005a*; *Wickiser et al., 2005b*), which should favor the terminator configuration. By mutating the U nucleotides at positions 46–48, we prevent constructs from becoming thermodynamically trapped in the terminator configuration during in-line probing, which

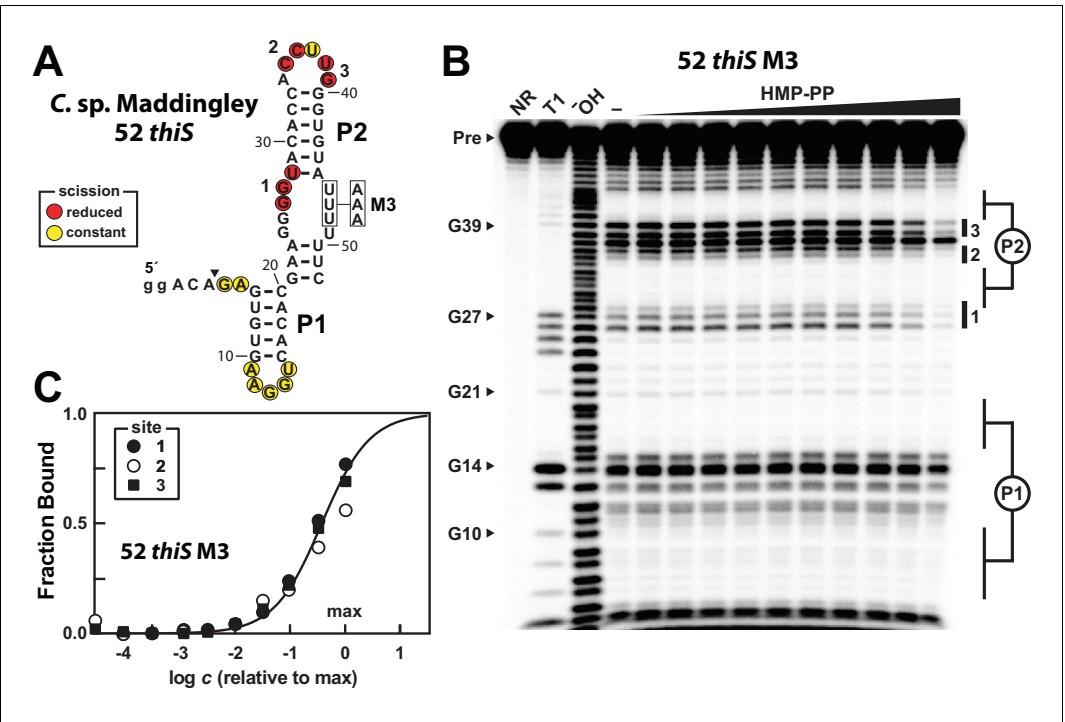

**Figure 5.** A modified representative HMP-PP riboswitch aptamer binds HMP-PP. (**A**) Sequence and secondary structure model of the 52 *thiS* RNA construct derived from the *thiS* gene of *C.* sp. Maddingley, and its M3 mutant (U46A, U47A, U48A). Both the WT and M3 constructs are known to retain gene control activity in response to ligand (*Figure 2*). Nucleotides at the 5′ end denoted in lowercase letters were added to enhance the efficiency of transcription in vitro. The characteristics of sites of spontaneous RNA cleavage resulting from in-line probing reactions in the absence or presence of HMP-PP, as derived from the image in B, are indicated. (**B**) PAGE analysis of 5′ $^{32}$P-radiolabeled 52 *thiS* M3 RNA that was subjected to in-line probing assays in the absence (–) or presence of HMP-PP. The ligand HMP-PP was enzymatically produced using ThiD protein and used in a range of concentrations relative to the maximum HMP-PP concentration (precise concentration unknown, see Materials and methods). The lane denoted "–" was supplemented with the processed reaction mixture including all the reagents (except ThiD protein) required for the preparation of HMP-PP. Thus, this lane includes ~0.5 mM HMP. NR, T1 and ⁻OH indicate no reaction, RNase T1 partial digestion (cleavage after G nucleotides), and alkaline-mediated partial digestion (cleaves after each nucleotide), respectively. Bands corresponding to precursor RNA (Pre) and enzymatic cleavage after certain G residues are annotated. Notable sites of ligand-mediated structural modulation are numbered 1 through 3. (**C**) Plot depicting the fraction of RNA bound to ligand versus the logarithm of the concentration of HMP-PP relative to the maximum (max) ligand concentration tested. The data points were generated by quantifying band intensity changes at sites 1, 2 and 3 in B with correction for loading and yield differences, including the general reduction in RNA cleavage observed in the final lane containing the maximum HMP-PP concentration tested. The line presents a theoretical 1:1 binding curve (Hill coefficient of 1) and is overlaid on the data points for comparison.

DOI: https://doi.org/10.7554/eLife.45210.010

The following figure supplements are available for figure 5:

**Figure supplement 1.** Mutational analysis of an HMP-PP aptamer.

DOI: https://doi.org/10.7554/eLife.45210.011

**Figure supplement 2.** HMP-PP mediated conformational rearrangement of construct 53 *thiS* M3 and related RNAs.

DOI: https://doi.org/10.7554/eLife.45210.012

permits HMP-PP binding to be observed. This same effect could be achieved with natural *thiS* motif sequences by having the U nucleotides at positions 46–48 remain sequestered in the RNA-exit channel (*Vassylyev et al., 2002*; *Hein et al., 2014*) of an RNA polymerase that is paused within the run of U nucleotides of the intrinsic terminator stem. We speculate that momentary ligand binding

during this paused state will be sufficient to prevent terminator formation on the vastly shorter time scale that is relevant to the genetic decision process in cells.

Overall, our biochemical data are strongly consistent with the hypothesis that *thiS* motif RNAs function as riboswitches that directly bind to HMP-PP. As a result, we favor renaming *thiS* motif RNAs as HMP-PP riboswitches. The pursuit of more precise biochemical and biophysical characteristics, and further evidence for the mechanism proposed above, will require both the judicious use of RNA constructs that form the aptamer configuration, and either the preparation of pure samples of HMP-PP with consideration for its relative instability or the use of more stable analogs that can trigger riboswitch function.

## Tandem TPP and HMP-PP riboswitches function as Boolean genetic logic gates

As noted above, approximately 30% of the representatives of HMP-PP riboswitches reside in tandem with TPP riboswitches. In these tandem systems, the TPP riboswitch always occurs first, and the TPP aptamer is always associated with its own expression platform, which is routinely a readily recognizable intrinsic terminator stem. The HMP-PP riboswitch follows each complete TPP riboswitch, and by its unique architecture, carries its own intrinsic terminator expression platform.

This arrangement, as observed in a representative from the bacterium *Clostridium lundense* (*Figure 6A*), makes apparent the genetic decisions made by the host. Specifically, abundant TPP should repress expression of the associated gene, which usually codes for a protein needed to biosynthesize HET-P (*Figure 1B*). This repression, regardless of the concentration of HMP-PP, makes sense because cells do not need to make more TPP when this enzyme cofactor is already abundant. However, when TPP is in short supply, transcripts should read through the first intrinsic terminator stem to begin transcription of the HMP-PP riboswitch. The HMP-PP aptamer only permits transcription read-through of its co-resident intrinsic terminator stem if the relative abundance of its ligand is high. If HMP-PP is abundant, then the resulting full-length mRNAs will produce proteins that biosynthesize more HET-P. This two-input tandem riboswitch system has a truth table that matches a 'converse nonimplication' Boolean logic function (*Figure 6B*).

To confirm that a tandem TPP and HMP-PP riboswitch system functions with this Boolean logic, we created a tandem riboswitch-reporter fusion construct based on the natural *C. lundense* RNA (*Figure 6A*), and examined its function in *B. subtilis* cells. Unfortunately, we cannot easily create cellular conditions to fully examine all four possible ligand states represented in the truth table (*Figure 6B*). Most obviously, TPP is an essential cofactor, and thus we cannot deplete its concentration to zero while maintaining cell viability. Also, the various TPP biosynthetic intermediates are likely to be present in low concentrations if the cell is continuously making more TPP, even at low flux. As a result, a cell might experience conditions where an individual nascent RNA transcript of the tandem riboswitch system could respond to any of the four possible states, but these responses when summed over multiple nascent transcripts will yield a hybrid gene expression response that generally reflects the state of ligands in the cell.

To fully examine the gene expression outputs for all four possible states, despite the complications described above, we only partially manipulated the state of ligands, while also creating artificial representations of the missing states by using riboswitch aptamer mutations. Specifically, by adding HMP to the culture medium, we could change the ligand state from state 2 (+TPP, – HMP-PP) to state 4 (+TPP,+HMP-PP). As predicted by the truth table, ligand states 2 and 4 do not yield expression of the reporter gene (*Figure 6C*). We also artificially created states 1 (– TPP, – HMP-PP) and 3 (– TPP,+HMP-PP) by employing disruptive mutations in the aptamers for TPP (M10), HMP-PP (M1), or both (M11). Only the artificial state 3 condition, as created by mutating the TPP aptamer and growing cells in the presence of HMP, yields robust reporter gene expression (*Figure 6C, M10*). This gene expression effect is lost when disabling mutations are placed in both aptamers (M11) to simulate state 1. These findings are consistent with our hypothesis that each tandem TPP and HMP-PP riboswitch system functions as a Boolean converse nonimplication logic gate.

## Discussion

All of our findings derived from bioinformatic, genetic and biochemical analyses indicate that *thiS* motif RNAs represent an unusual class of riboswitches that sense and respond to the thiamin

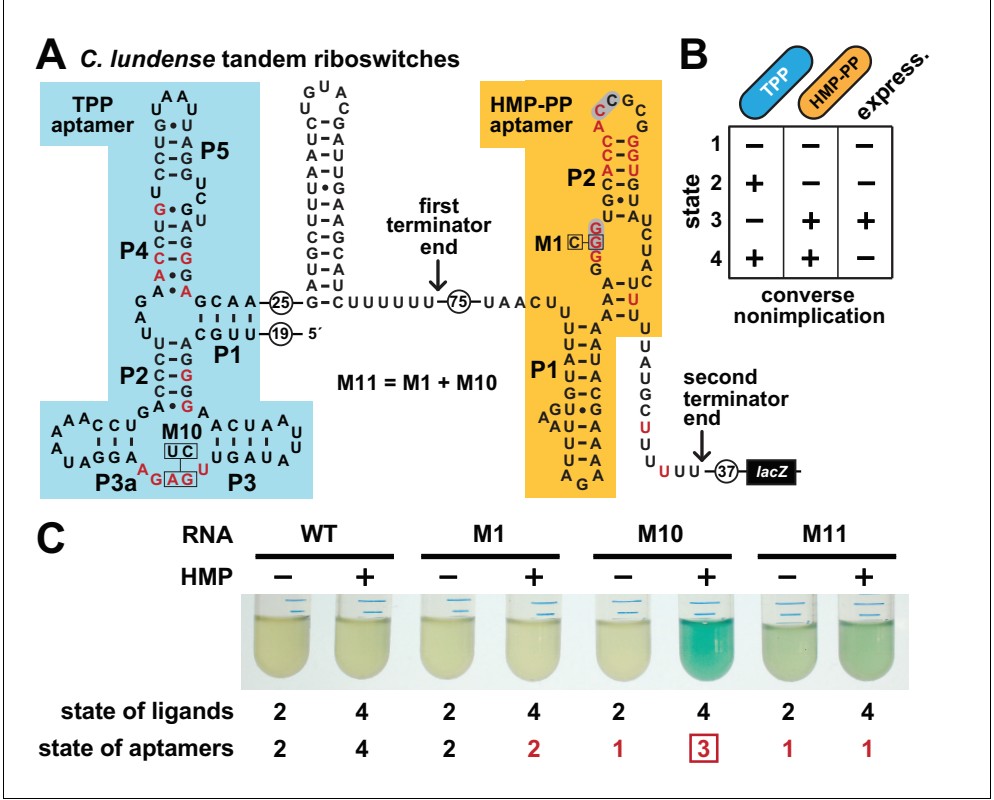

**Figure 6.** Tandem TPP and HMP-PP riboswitches form a two-input Boolean logic gate to regulate gene expression. (**A**) Sequence and secondary structure of a tandem riboswitch arrangement derived from the *thiW* gene from *C. lundense* DSM 17049 that is responsive to TPP and HMP-PP ligands. Red letters identify positions matching the most highly conserved nucleotides in the consensus sequence models for TPP (*McCown et al., 2017*) and HMP-PP (*Figure 1A*) aptamers. (**B**) Predicted truth table for tandem TPP and HMP-PP riboswitches represents a converse nonimplication logic gate. Minus and plus symbols, respectively, represent low or high gene expression (express.) or ligand concentrations as indicated. (**C**) Reporter gene expression of a *B. subtilis* strain expressing ThiD, and carrying either a wild-type (WT) or a mutant (M1, M10 or M11) *C. lundense* tandem riboswitch-*lacZ* reporter fusion construct as described in A. Cells were grown in rich (LB) medium containing either no (–) or 10 µM additional HMP and 200 µg mL$^{-1}$ X-gal. The numbers listed as 'state of ligands' represents the cellular presence or absence of ligands corresponding to the 'states' depicted in the truth table in B. The numbers listed as 'state of aptamers' represents the expected status (free or ligand bound) of the aptamers in the riboswitch, again as depicted in the truth table. Red numbers identify states artificially created by riboswitch mutations. The box identifies state 3, which is created by the addition of HMP to construct M10.
DOI: https://doi.org/10.7554/eLife.45210.013

pyrophosphate biosynthetic intermediate HMP-PP. Its architecture is unusual because the majority of the nucleotides and structures required for ligand recognition appear to be encompassed by an intrinsic transcription terminator, which forms in a mutually-exclusive manner to regulate gene expression. All other riboswitch classes reported to date (*McCown et al., 2017*) that employ an intrinsic terminator structure as an expression platform carry the most-highly conserved nucleotides and structural features largely upstream of, and sometimes only partly overlapping, the terminator structure. This unique overlapping architecture of HMP-PP riboswitches provides bacteria with a highly-compact, yet effective RNA-based gene control system for measuring HMP-PP concentrations and tuning the production of HET-P to more efficiently produce TPP.

The embedded nature of the aptamer and expression platform domains, however, complicated the pursuit of experiments typically used to biochemically validate a newly discovered riboswitch class. Most importantly, our experiments required the use of non-native constructs to successfully reveal HMP-PP binding in vitro. Therefore, it is notable that modifications to certain key constructs maintain function in vitro or in vivo. For example, the trimming of nucleotides at the 3′ region of the

terminator stem to create RNA construct 52 *thiS* yields a molecule that is expected to represent the exposed portion of a polymerase-paused nascent RNA transcript (*Figure 4A*). This truncated construct adopts the predicted P1 stem of the aptamer configuration (*Figure 4B*). Also, the addition of nucleotide changes to create the M3 variant of 52 *thiS* yields a construct that binds HMP-PP, whereas this same M3 mutation in a related construct retains ligand-dependent gene-control function in cells (*Figure 2C*).

Our results also reveal a mechanism for gene control involving the stalling of RNA polymerase at a precise location to yield an exposed nascent RNA transcript that is transiently capable of forming an HMP-PP binding pocket. The 52 *thiS* construct predominantly exists in the aptamer configuration, which is distinct from even modestly longer forms that partially (53 *thiS*) or exclusively (54 *thiS*) adopt the terminator configuration (*Figure 4*). If HMP-PP is present and docks in the aptamer when the nascent mRNA transcript is exposed up to nucleotide position 52, terminator stem formation is precluded and transcription proceeds to generate the full-length RNA. Forward movement of RNA polymerase by even a single nucleotide yields a construct that begins to commit to the terminator stem structure, and at this point transcription would be terminated regardless of the presence of HMP-PP.

The *thiS* motif was originally identified as a riboswitch candidate by using an updated computational pipeline (*Stav et al., 2019*) that focuses attention on relatively long, GC-rich IGRs. This method of riboswitch discovery was developed to improve our ability to uncover rare and/or small riboswitch candidates. Thousands of riboswitch classes are predicted to remain undiscovered in bacteria (*Ames and Breaker, 2010*; *McCown et al., 2017*), and many of these are likely to be exceedingly rare and/or difficult to find because of their structural simplicity. Experimental demonstration that *thiS* motif RNAs are representatives of a compact riboswitch class that responds to HMP-PP provides initial confirmation that the updated computational pipeline can uncover riboswitch classes that have resisted discovery by other bioinformatic or genetic methods. We are hopeful that many more riboswitch classes will be revealed by employing such bioinformatics search algorithms on additional sequenced bacterial genomes.

Finally, we note that the discovery of an HMP-PP riboswitch provides yet another example of a modern RNA class that selectively binds a molecule that was likely present in an RNA World (*Benner et al., 1989*; *Gilbert, 1986*). Many of the common enzyme cofactors are derived from RNA nucleotides or their precursors, including TPP and its precursor HMP-PP, which is a characteristic previously used to support the hypothesis that these molecules predate proteins (*White, 1976*). TPP and many other RNA-derived coenzymes might have participated in a complex metabolic state run entirely by RNA enzymes and receptors, and therefore many riboswitches for coenzymes likewise might be molecular descendants from RNAs that utilized these coenzymes long ago (*Breaker, 2012*; *McCown et al., 2017*). If true, HMP-PP riboswitches provide another opportunity to learn how simple RNA sequences and structures could have selectively bound important nucleotide-derived metabolites in primitive organisms of the RNA World.

## Materials and methods

**Key resources table**

| Reagent type (species) or resource | Designation | Source or reference | Identifiers | Additional information |
|---|---|---|---|---|
| Gene (*Salmonella typhimurium*) | *thiD* | NA | GenBank: AZI82914.1 | |
| Strain, strain background (*Bacillus subtilis*) | *B. subtilis* strain 168 | Bacillus Genetic Stock Center | 1A1 | |
| Strain, strain background (*Escherichia coli*) | BL21 (DE3) | New England Biolabs | C2527I | |
| Genetic reagent (*Bacillus subtilis*) | Δ*thiS* | Bacillus Genetic Stock Center | 11680 | |
| Genetic reagent (*Bacillus subtilis*) | Δ*thiD* | Bacillus Genetic Stock Center | 11710 | |

*Continued on next page*

*Continued*

| Reagent type (species) or resource | Designation | Source or reference | Identifiers | Additional information |
|---|---|---|---|---|
| Genetic reagent (*Bacillus subtilis*) | Δ*thiE* | Bacillus Genetic Stock Center | 38290 | |
| Recombinant DNA reagent | pDG1661-*thiS*-*lacZ* | This paper | | HMP-PP riboswitch β-galactosidase reporter construct |
| Recombinant DNA reagent | pDG1661-tandem-lacZ | This paper | | Tandem TPP and HMP-PP riboswitch β-galactosidase reporter construct |
| Recombinant DNA reagent | pDG148-Stu-*thiD* | This paper | | pDG148-Stu expression vector for ThiD protein |
| Recombinant DNA reagent | pETDuet:His-ThiD | This paper | | pETDuet expression vector for N-terminal HisX6 tagged ThiD protein |
| Commercial assay or kit | *E. coli* RNA polymerase holoenzyme | New England Biolabs | M0551S | |
| Chemical compound, drug | (4-amino-2-methylpyrimidin-5-yl)methanol | Enamine Ltd | EN300-135390 | |
| Chemical compound, drug | X-Gal | MilliporeSigma | 3117073001 | |
| Chemical compound, drug | 4-Methylumbelliferyl β-D-galactopyranoside | MilliporeSigma | M1633 | |
| Chemical compound, drug | $[\gamma\text{-}^{32}P]$-ATP | PerkinElmer | | |
| Chemical compound, drug | $[\alpha\text{-}^{32}P]$-ATP | PerkinElmer | | |
| Software, algorithm | GraphPad Prism (RRID:SCR_002798) | GraphPad Prism (https://graphpad.com) | | Version 7 |

## Chemicals, oligonucleotides and bacterial strains

Chemicals were purchased from Sigma-Aldrich with the exception of 4-amino-5-hydroxymethyl-2-methylpyrimidine, also known as (4-amino-2-methylpyrimidin-5-yl) methanol, which was purchased from Enamine Ltd. The radiolabeled molecules $[\gamma\text{-}^{32}P]$-ATP and $[\alpha\text{-}^{32}P]$-ATP were purchased from PerkinElmer. All enzymes were purchased from New England BioLabs unless otherwise specified. Synthetic DNA oligonucleotides were purchased from Sigma-Aldrich or Integrated DNA Technologies. A list of oligonucleotides used in this study can be found in *Supplementary file 3*.

BL21 (DE3) *E. coli* cells were purchased from New England Biolabs and transformed with the appropriate plasmid for overproduction of the ThiD protein, whose enzymatic function ultimately was confirmed by monitoring the production of HMP-PP by mass spec analysis. The parent *B. subtilis* 168 (BGSC 1A1) strain, and the corresponding mutant strains Δ*thiS* (BGSC 11680), Δ*thiE* (BGSC 38290) and Δ*thiD* (BGSC 11710) were obtained from the Bacillus Genetic Stock Center (BGSC) at The Ohio State University, and genetically modified as described herein. All bacterial strains were verified by testing for the expected growth, antibiotic resistance, and reporter gene expression phenotypes.

## Bioinformatic analysis

Representative *thiS* motif RNAs used to create the consensus sequence and structure models (*Figure 1A*) were identified using Infernal 1.1 (*Nawrocki and Eddy, 2013*) from RefSeq version 80 and certain environmental microbial databases as described previously (*Weinberg et al., 2017*). A total of 400 unique and complete representatives were used to generate an updated consensus model relative to that published previously (*Stav et al., 2019*). The consensus sequence and structural models were derived by using the R2R algorithm (*Weinberg and Breaker, 2011*), which

employs weighting. To prevent irrelevant differences between the aptamer and terminator configurations caused by weighting, the consensus model of the aptamer configuration was used to annotate the consensus sequence for the terminator configuration for the overlapping region.

## Riboswitch reporter assays

Riboswitch-reporter constructs, integrating riboswitch representatives from either *C. sp* Maddingley or *C. lundense* DSM 17049, were prepared as synthetic oligonucleotides, amplified by PCR and cloned into vector pDG1661 upstream of the *E. coli lacZ* gene as described previously (*Sudarsan et al., 2003*; *Nelson et al., 2017*). Transcription initiation of the constructs is driven by the *B. subtilis lysC* gene promoter. The resulting WT and mutant reporter constructs were integrated into the *amyE* locus of WT (1A1 strain 168 Δ*trp)* or thiamin biosynthetic knockout strains (Δ*thiS*, Δ*thiE* or Δ*thiD*) as indicated. The resulting transformed strains were verified as previously described (*Sherlock et al., 2018*).

The *thiD* gene construct was generated by amplifying this gene from *B. subtilis* genomic DNA by PCR and inserted into the StuI site of a modified pDG148 vector using ligation-independent cloning as described previously (*Joseph et al., 2001*). The *lacI* gene in this vector has been mutated so that the *thiD* gene is expected to give constitutive expression. The resulting protein expression vector was then transformed into *B. subtilis* strains containing WT or mutant riboswitch reporter constructs, as indicated for each experiment.

Riboswitch-reporter assays were performed by inoculating various *B. subtilis* strains into Lysogeny Broth (LB) with appropriate antibiotics and growing overnight at 37°C. For liquid-culture reporter assays with thiamin biosynthetic knock-out strains, overnight cultures grown in LB were then diluted 1/20 into Spizizen glucose minimal medium (GMM) (*Anagnostopoulos and Spizizen, 1961*) and grown overnight at 37°C. The residual thiamin from LB is sufficient for growth in GMM over the duration of the assay. For riboswitch reporter experiments with ThiD-producing strains, bacteria were diluted directly into LB and grown overnight with or without supplementation with HMP. Liquid media (LB or GMM) was supplemented with X-gal (200 µg mL$^{-1}$) to allow visual detection of reporter gene expression. Similarly, reporter expression analysis using 4-methylumbelliferyl β-D-galactopyranoside was conducted as described previously (*Nelson et al., 2015*; *Atilho et al., 2019*) to establish fluorescence units.

## Expression and purification of ThiD

An N-terminal 6xHis-tagged *thiD* gene from *Salmonella typhimurium* was cloned into a pETDuet vector, which was then transformed into *E. coli* strain BL21(DE3). Transformed cells were grown in Terrific Broth medium until the $OD_{600}$ reached 0.8. The resulting culture was incubated overnight at 16°C for protein expression that was induced by the addition of 0.3 mM isopropyl β-D-1-thiogalactopyranoside. Cells were pelleted, resuspended in Buffer A [50 mM Tris (pH 8 at 23°C), 400 mM NaCl, 10 mM imidazole, 5% glycerol, 0.1 mM tris(2-carboxyethyl)phosphine (TCEP)], and lysed with a microfluidizer. The resulting lysate was clarified via centrifugation, applied to NiNTA resin, and washed with 10 column volumes of Buffer A. Tagged protein was eluted from the column with three column volumes of Buffer B [50 mM Tris (pH 8 at 23°C), 400 mM NaCl, 400 mM imidazole, 5% glycerol, 0.1 mM TCEP] and applied to a size-exclusion column equilibrated in Buffer C [50 mM Tris (pH 8 at 23°C), 150 mM NaCl, 0.1 mM TCEP]. 10% Glycerol (v/v, final concentration) was added to the protein sample before storage at −80°C.

## Enzymatic preparation of HMP-PP

Pyrophosphorylation of HMP was carried out enzymatically as described previously (*Hanes et al., 2007*). Briefly, the reaction was initiated by adding 10 µL of *Salmonella typhimurium* HMP-P kinase (ThiD, 20 mg/mL) to a 90 µL reaction preparation to yield a final concentration of 5 mM HMP, 20 mM ATP, 50 mM Tris-HCl buffer (pH 7.5 at 23°C), 2 mM TCEP, and 5 mM $MgCl_2$. This 100 µL reaction mixture was allowed to incubate at 23°C overnight, at which time the protein was removed by ultrafiltration using an Amicon Ultra-0.5 centrifugal filter unit with a 3 kDa cutoff membrane. The HMP-PP generated in this manner was used without further purification, typically on the same day but no later than 2 weeks after production. HMP-PP stock solutions were stored at −20°C. Control assays for transcription termination and in-line probing conducted without the addition of

enzymatically prepared HMP-PP contained an equivalent amount of the enzymatic reaction preparation wherein the ThiD protein was excluded. These control assays therefore contain HMP, whereas test reactions contain HMP-PP.

## Mass spectrum analysis of HMP-PP

Enzymatically prepared HMP-PP samples were sent to the MS and Proteomics Resource at Yale University for analysis. The presence of HMP-PP was confirmed by data (*Figure 3C—figure supplement 1*) generated using a Thermo Scientific LTQ Orbitrap ELITE mass spectrometer. Data was acquired and analyzed with Xcalibur (v2.1). Peaks were considered to have the same mass-to-charge ratio as HMP-PP if they were within 10 ppm of the calculated ratio.

## In vitro transcription termination assays

The protocol used for single-round in vitro transcription assays was adapted from that described previously (*Landick et al., 1996*). DNA constructs were designed to include the promoter sequence of the *lysC* gene from *B. subtilis*, the riboswitch aptamer, and the expression platform of the *thiS* gene from *C.* sp Maddignly to 33 nucleotides following the terminator stem. Additional non-native nucleotides were added to the 5′ region upstream of the HMP-PP aptamer to increase the amount of [α-$^{32}$P]-ATP incorporation.

To assemble each in vitro transcription reaction, approximately 2 pmol of the purified, PCR amplified DNA template was added to a transcription initiation mixture [final concentration of 20 mM Tris (pH 8.0 at 23°C), 20 mM NaCl, 14 mM MgCl$_2$, 100 μM EDTA, 10 μg mL$^{-1}$ bovine serum albumin, 130 μM ApA dinucleotide, 1% glycerol, 0.04 U μL$^{-1}$ *E. coli* RNA polymerase holoenzyme, 2.5 μM GTP, 2.5 μM UTP, and 1 μM ATP]. Approximately 8 μCi [α-$^{32}$P]-ATP was added to the 90 μL transcription reaction and transcription was allowed to proceed at 37°C for 30 min, leading to formation of a stalled polymerase complex before the first cytidine of each transcript. The reaction mixture was then distributed in 8 μL aliquots into separate microfuge tubes, which contained 1 μL of an HMP or HMP-PP solution plus 1 μL of 10x elongation buffer (200 mM Tris [pH 8.0 at 23°C], 200 mM NaCl, 140 mM MgCl$_2$, 1 mM EDTA, 1 mg mL$^{-1}$ heparin, 1.5 mM each of ATP, GTP, and CTP, and 0.5 mM UTP). For transcription termination assays, the maximum concentration of HMP-PP is 1.5 mM, assuming 100% enzymatic conversion of HMP and no loss to instability. However, HMP-PP concentration is likely substantially lower due to incomplete conversion of HMP to HMP-PP (*Figure 3—figure supplement 1*). Transcription elongation was allowed to proceed for 45 min at 37°C.

The transcription products were separated by denaturing (8 M urea) 10% polyacrylamide gel electrophoresis (PAGE) then imaged and quantified using a Typhoon Phosphorimager and ImageQuaNT software. The fraction of full length (FL) and terminated (T) RNA transcripts was calculated by measuring band intensity values and using the equation Fraction FL = (FL intensity)/(FL intensity +T intensity). The differences in specific activities between the FL and T products due to [α-$^{32}$P]-ATP incorporation were considered negligible.

## RNA oligonucleotide preparation

RNAs were prepared by in vitro transcription using DNA oligonucleotides containing a T7 RNA polymerase promoter sequence upstream of the desired template sequence. The resulting desired RNA transcripts were purified, enzymatically 5′ $^{32}$P-labeled, and repurified as previously described (*Mirihana Arachchilage et al., 2018*; *Atilho et al., 2019*).

## RNA in-line probing analysis

In-line probing assays (*Soukup and Breaker, 1999*; *Regulski and Breaker, 2008*) were performed precisely as described previously (*Atilho et al., 2019*; *Mirihana Arachchilage et al., 2018*). The maximum HMP-PP concentration (max) is achieved by using a 3/10 dilution of enzymatically prepared HMP-PP as described above. The control reaction was performed using the 1/10 dilution of the control sample that lacks HMP-PP.

## Acknowledgements

We thank Adam Roth, Narasimhan Sudarsan, Shira Stav, and other members of the Breaker laboratory for helpful discussions. We are grateful to Rob Bjornson for assisting our use of the Yale Life Sciences High Performance Computing Center (NIH grant RR19895-02), Dr. Yong Xiong for assistance in the production and purification of the ThiD protein, and WeiWei Wang for assistance with mass spectrometry. RMA was supported by the National Science Foundation Graduate Research Fellowship Program (DGE1122492). This work was also supported by NIH grants to RRB (GM022778). In addition, RRB is supported by the Howard Hughes Medical Institute.

## Additional information

### Funding

| Funder | Grant reference number | Author |
|---|---|---|
| National Institutes of Health | P01 GM022778 | Ronald R Breaker |
| National Science Foundation | Graduate Student Fellowship | Ruben M Atilho |
| Howard Hughes Medical Institute | Investigator Funding | Ronald R Breaker |

The funders had no role in study design, data collection and interpretation, or the decision to submit the work for publication.

### Author contributions

Ruben M Atilho, Conceptualization, Data curation, Formal analysis, Validation, Investigation, Visualization, Methodology, Writing—review and editing; Gayan Mirihana Arachchilage, Conceptualization, Data curation, Formal analysis, Methodology, Writing—review and editing; Etienne B Greenlee, Data curation, Software, Investigation, Writing—review and editing; Kirsten M Knecht, Methodology, Writing—original draft, Writing—review and editing; Ronald R Breaker, Conceptualization, Supervision, Funding acquisition, Investigation, Methodology, Writing—original draft, Project administration, Writing—review and editing

### Author ORCIDs

Ronald R Breaker http://orcid.org/0000-0002-2165-536X

### Decision letter and Author response

Decision letter https://doi.org/10.7554/eLife.45210.019
Author response https://doi.org/10.7554/eLife.45210.020

## Additional files

### Supplementary files

• Supplementary file 1. This file provides the coordinates for each HMP-PP riboswitch representative, and includes the genes associated with each representative when such annotations are available.
DOI: https://doi.org/10.7554/eLife.45210.014

• Supplementary file 2. This file presents sequence alignments in Stockholm format for individual HMP-PP riboswitch aptamers and for tandem TPP and HMP-PP aptamers.
DOI: https://doi.org/10.7554/eLife.45210.015

• Supplementary file 3. This file includes four supplementary data figures and a table listing the DNA constructs used in the study.
DOI: https://doi.org/10.7554/eLife.45210.016

• Transparent reporting form
DOI: https://doi.org/10.7554/eLife.45210.017

**Data availability**

All data generated or analyzed during this study are included in the manuscript and supporting files.

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
