## [Decision Letter]

Thank you for submitting your article "A bacterial riboswitch class for the thiamin precursor HMP-PP employs a terminator-embedded aptamer" for consideration by *eLife*. Your article has been reviewed by four peer reviewers, one of whom is a guest Reviewing Editor, and the evaluation has been overseen Gisela Storz as the Senior Editor. The following individual involved in review of your submission has agreed to reveal their identity: Jinwei Zhang (Reviewer #4).

The reviewers have discussed the reviews with one another and the Reviewing Editor has drafted this decision to help you prepare a revised submission.

Summary:

The manuscript by Atilho et al. presents a new twist in the riboswitch field: an unexpected finding of a ligand-sensing domain embedded in the expression platform, a transcriptional terminator. It begs the question: how many of these mini-systems are hidden out there? The authors did excellent work proving that *thiS* motif is a functional riboswitch that responds to an important precursor of thiamine pyrophosphate, the active form of vitamin B1. The size of this riboswitch is remarkably small, and yet not only can this riboswitch recognizes the ligand very specifically, it can also function in conjunction with another riboswitch to precisely balance the expression of relevant genes. Undoubtedly, this work will attract attention of many researchers in different fields and will force researchers to re-think composition and role of one of the most widely-spread molecular elements – transcription terminator. The results will be of great interest to broad readership of *eLife*. Collectively, the reviewers have a few major points for clarification.

Essential revisions:

(The authors do not necessarily need to address every question raised here, however, an effort to provide a more completely picture of the *thiS-thiM* regulated bacterial physiology would certainly further improve the impact of the work.)

1) There is a lack of sufficient information on the distribution and genetic arrangements of *thiS* riboswitches. The authors cited an unpublished paper by Stav et al., 2019 but did not provide this manuscript and did not reveal details relevant to the current work. It is unclear when the Stav et al. work will be published and it is recommended to give a better summary of the findings and perhaps a supplementary figure with genetic arrangements, which would nicely complement a supplementary figure with riboswitch sequences. What are "uncommon" genes that reside downstream of *thiS* motif? Are all of them mentioned in the third paragraph of the Introduction? Can the assigned specificity of this motif explain regulation of these uncommon genes? If no, could it be possible for the riboswitch to recognize more than one ligand, especially given the fact that HMP-PP binding may be weak? Can this motif be a family of distinct riboswitches? Were there any attempts made to separate the motif on submotifs based on the genetic association? What are transporter genes controlled by the motif? *ThiW* appears to be an essential thiazole transporter and although the presence of the *thiS* motif controlling this protein suggests the regulation of coupling of the two TPP moieties, one can imagine direct regulation of a subtype of this motif by precursors of HET-P (Figure 1B). Figure 1B shows several genes associated with this motif and it would be helpful to identify and discuss these genes. Is each of these genes preceded by its own copy of the *thiS* motif? Are these genes controlled by a single thiS motif and not by a tandem *this-thiM* motif? The third paragraph of the subsection “The unusual architecture and genetic distribution of the *thiS* riboswitch candidate”, says that 30% of the known representatives reside near TPP riboswitches. Is there a genetic bias for the tandem riboswitch arrangement? Why the consensus motif (Figure 1A) is presented differently in the terminator and aptamer configurations? In the aptamer configuration, the loop in P1 is shown as variable while no extra nts are indicated for this region in the terminator configuration. The variable loop size contradicts the statement: "This structure exhibits all the features characteristic of bacterial intrinsic terminator stems, including an uninterrupted and strong base-paired stem". Actually the entire 5' end is different in the two configurations. Answers to these questions will help to convince this reviewer that the *thiS* motif is indeed a HMP-PP binder and will also help microbiologists to better understanding regulatory networks associated with vitamin production in bacteria.

2) The novel computational pipeline the authors describe is from unpublished work. Though not the subject of this manuscript, it is hard to see how this new approach differs from approaches used by this lab in the past. What makes this new approach more sensitive to rare riboswitch elements? Does this sensitivity arise from looking at smaller groups of bacteria (5 species) instead of across numerous species? The novelty of the approach is touted throughout, but not well explained.

---

## [Author Response]

1) There is a lack of sufficient information on the distribution and genetic arrangements of thiS riboswitches. The authors cited an unpublished paper by Stav et al., 2019 but did not provide this manuscript and did not reveal details relevant to the current work. It is unclear when the Stav et al. work will be published and it is recommended to give a better summary of the findings and perhaps a supplementary figure with genetic arrangements, which would nicely complement a supplementary figure with riboswitch sequences.What are "uncommon" genes that reside downstream of thiS motif? Are all of them mentioned in the third paragraph of the Introduction? Can the assigned specificity of this motif explain regulation of these uncommon genes? If no, could it be possible for the riboswitch to recognize more than one ligand, especially given the fact that HMP-PP binding may be weak? Can this motif be a family of distinct riboswitches? Were there any attempts made to separate the motif on submotifs based on the genetic association? What are transporter genes controlled by the motif? ThiW appears to be an essential thiazole transporter and although the presence of the thiS motif controlling this protein suggests the regulation of coupling of the two TPP moieties, one can imagine direct regulation of a subtype of this motif by precursors of HET-P (Figure 1B). Figure 1B shows several genes associated with this motif and it would be helpful to identify and discuss these genes. Is each of these genes preceded by its own copy of the thiS motif? Are these genes controlled by a single thiS motif and not by a tandem this-thiM motif? The third paragraph of the subsection “The unusual architecture and genetic distribution of the thiS riboswitch candidate”, says that 30% of the known representatives reside near TPP riboswitches. Is there a genetic bias for the tandem riboswitch arrangement? Why the consensus motif (Figure 1A) is presented differently in the terminator and aptamer configurations? In the aptamer configuration, the loop in P1 is shown as variable while no extra nts are indicated for this region in the terminator configuration. The variable loop size contradicts the statement: "This structure exhibits all the features characteristic of bacterial intrinsic terminator stems, including an uninterrupted and strong base-paired stem". Actually the entire 5' end is different in the two configurations. Answers to these questions will help to convince this reviewer that the thiS motif is indeed a HMP-PP binder and will also help microbiologists to better understanding regulatory networks associated with vitamin production in bacteria.

Point 1A: The reviewers note a lack of bioinformatics information on the *thiS* motif that is otherwise typically supplied for novel riboswitches.

As the reviewers correctly speculate, this comprehensive information set was included for publication in an earlier manuscript (Stav et al.) reporting the bioinformatics discovery of the *thiS* riboswitch candidate (and numerous other RNA motifs). Unfortunately, this manuscript had experienced a remarkably long delay in the initial review process (~9 months). On March 7, 2019 we received notice that this original discovery manuscript has been accepted for publication by BMC Microbiology. Regardless, we have also decided to include an updated and detailed bioinformatics dataset as a Supplementary file 1 to the current revised manuscript for *eLife*. The original two supplementary files have been renamed accordingly. See also our response to Major Point 2 below.

Point 1B: The reviewers encourage us to comment on uncommon gene associations for the *thiS* motif and to discuss the possibility that the riboswitch aptamer might recognize different ligands to regulate these uncommon genes.

This question from the reviewers is an important one, particularly given the recent and striking examples of variant “*ykkC* orphan riboswitch” RNAs that have adapted to respond to at least five distinct ligands. However, nearly all the genes associated with *thiS* motif RNAs appear to code for proteins that participate in the production of HET-P or its coupling to HMP-PP. In some instances, associated genes from organisms whose genomes have not been annotated appear as ‘proteins of unknown function’. However, we have examined these genes and nearly all of these can easily be classified as *thiS*. Therefore, we do not have any indication that *thiS* motif RNAs have changed their specificity to regulate genes unrelated to thiamin pyrophosphate biosynthesis in general, or HET-P biosynthesis in particular. Also, the sequence alignments are quite uniform (matching the consensus), and so we do not believe that the RNAs have specialized based on any particular gene association. To clarify these consistent characteristics, we have edited the following text to the paragraph noted by the reviewers:

“Most other genes associated with *thiS* motif RNAs appear to code for proteins that participate in the production of HET-P or its fusion to 4-amino-5-hydroxymethyl-2-methylpyrimidine diphosphate (HMP-PP), to ultimately produce the bioactive coenzyme thiamin pyrophosphate (TPP) (Jurgenson et al., 2009).”

Point 1C: The reviewers ask for specific clarification on the *thiS* motif association with a putative thiazole transporter.

We have found that transporter annotations can be uncertain (and are frequently incorrect) and so we must speculate with considerable caution. However, the transporter genes associated with *thiS* motif RNAs would make most sense if they import molecules related to HET-P production. Specifically, we believe that the *thiW* gene association is a logical association for an HMP-PP “ON” switch because excess HMP-PP would trigger the import of the thiazole derivative HET, which can be used to biosynthesize more HET-P. In other words, we see no reason to doubt the current annotation of this transporter gene, and thus we have no evidence of a ligand specificity switch based on the current transporter gene annotations associated with *thiS* motif representatives.

Point 1D: The reviewers encourage us to discuss each of the genes associated with *thiS* motif RNAs and to describe the arrangements of their associated riboswitches.

Generally, the *thiS* gene is the gene most proximal to the RNA motif, whereas the other genes appear to be part of an operon regulated by HMP-PP riboswitches. Based on these gene names, and these operon arrangements, we believe the gene products catalyze the reaction steps for HET-P or TPP biosynthesis as depicted in Figure 1B. This list of gene functions has now been included in the new Supplementary file 1 that reports the various *thiS* representatives and their associated genes. Unfortunately we do not have a basis for discussing these predicted functions beyond that implied by the figure.

Point 1E: The reviewers ask if there is any special bias in the distribution of tandem TPP and HMP-PP (*thiS*) riboswitch systems.

We do not observe any special bias in the distribution of single versus tandem riboswitches, and therefore we cannot make any profound conclusions regarding the use of these arrangements. Presumably some species are exploiting the advantage of using tandem TPP and HMP-PP riboswitches, whereas others fail to take advantage of this arrangement. Such imperfect exploitations of tandem riboswitch arrangements have been observed for other riboswitch classes in the past.

Point 1F: The reviewers ask why the consensus models between the two structural states depicted in Figure 1A are different.

The vast majority of the nucleotide annotations in the original Figure 1A were identical between the two models. However, sequence alignments that match the proposed distinct structural states (specifically the complete terminator stem versus P1) caused slight differences in the consensus sequence model near the 5’ terminus (due to alignment alterations). To avoid depicting these differences, we have revised Figure 1A so that we depict the two structural states using only the consensus nucleotide sequence model for the aptamer configuration alignment. This updated figure is included in the revised manuscript file.

Point 1G: The reviewers state that our claim that the motif has the classic characteristics of an intrinsic terminator stem is incorrect.

We disagree. As we indicate in the original manuscript, the structural model as drawn on the left side of Figure 1A (terminator configuration) has all the key characteristics expected for a classic intrinsic terminator. Specifically, this part of the figure depicts a long hairpin with strong and continuous base-pairing, followed by a run of U nucleotides. Even as originally drafted, we do not believe that readers will be unduly confused by the right side of Figure 1A (aptamer configuration), as we carefully label these two structural states. Thus, given the clarity of the text and the figure annotations, we feel that the original text and the revised graphics should serve the readers well, and so no additional edits beyond those noted in Point 1F have been made.

2) The novel computational pipeline the authors describe is from unpublished work. Though not the subject of this manuscript, it is hard to see how this new approach differs from approaches used by this lab in the past. What makes this new approach more sensitive to rare riboswitch elements? Does this sensitivity arise from looking at smaller groups of bacteria (5 species) instead of across numerous species? The novelty of the approach is touted throughout, but not well explained.

The reviewers ask us to clarify the differences between the current discovery pipeline used to identify *thiS* RNAs compared to our previously used methods. Specifically, how is the current pipeline able to uncover rare riboswitch candidates?

These questions on the bioinformatics pipeline are fully addressed in the preceding manuscript (Stav et al.) that is mentioned in the response to Major Point 1A above. Briefly, our approach is a substantially updated version of a search strategy we first published in 2009. Specifically, this approach enriches for stretches of bacterial genomes that are likely to carry novel noncoding RNA motifs like riboswitches by collecting long, GC-rich intergenic regions (IGRs). This massively reduces the number of false positive candidates, and allows us to focus on motifs that are more likely to have the biochemical functions we seek. Although the delay in the initial review of the Stav et al. manuscript has been unusually long, as noted in the response to Point 1A, we have recently received word that our revised manuscript has been accepted for publication.